# Research on Coated Tool Life and Wear in Ta-2.5W Alloy Turning

**DOI:** 10.3390/ma17071481

**Published:** 2024-03-24

**Authors:** Bo Hu, Zhengqing Liu, Yang Wu, Qiucheng Wang, Dayu Shu

**Affiliations:** 1College of Mechanical Engineering, Zhejiang University of Technology, Hangzhou 310023, China; 2112102194@zjut.edu.cn (B.H.); liuzhengqing@zjut.edu.cn (Z.L.); 2Southwest Technology and Engineering Research Institute, Chongqing 400039, China; cquwuyang@163.com (Y.W.); shudayu1980@163.com (D.S.)

**Keywords:** Ta-2.5W alloy, coated tool, turning, cutting temperature, tool life, tool wear

## Abstract

Due to its inherent high hardness, strength, and plasticity, tantalum–tungsten (Ta-W) alloy poses a considerable challenge in machining, resulting in pronounced tool wear, diminished tool lifespan, and suboptimal surface quality. This study undertook experiments utilizing uncoated carbide tools, TiAlN-coated carbide tools, and AlTiN-coated carbide tools for machining Ta-2.5W alloy. The investigation delved into the intricacies of surface temperature, tool longevity, and the distinctive wear characteristics under varying coating materials and cutting parameters. Concurrently, a comprehensive exploration of the wear mechanisms affecting the tools was conducted. Among the observed wear modes, flank wear emerged as the predominant issue for turning tools. Across all three tool types, adhesive wear and diffusion wear were identified as the principal wear mechanisms, with the TiAlN-coated tools displaying a reduced level of wear compared to their AlTiN-coated counterparts. The experimental findings conclusively revealed that TiAlN-coated carbide tools exhibited an extended tool lifespan in comparison to uncoated carbide tools and AlTiN-coated carbide tools, signifying superior cutting performance.

## 1. Introduction

Tantalum–tungsten (Ta-W) alloy stands out for its exceptional physical and chemical properties, including a high melting point, elevated density, impressive corrosion resistance, and remarkable toughness. Consequently, it finds widespread applications across diverse fields such as chemical engineering [1], electronics [2], and aerospace [3], and as a crucial material for warhead components in weapons systems [4]. However, due to its inherent characteristics such as plasticity and work hardening, the machining of Ta-2.5W is often challenging. The machining process often grapples with issues like elevated cutting temperatures, substantial cutting forces, and complications in chip formation. Subsequent tool wear and chip formation can lead to the deformation of cutting edges, resulting in rapid tool failure and, consequently, poor machined surface quality [5,6,7]. Therefore, a comprehensive exploration of tool wear during the turning of Ta-2.5W is imperative.

The effective application of cutting fluids serves to diminish the friction coefficient between the tool and workpiece, thereby minimizing tool wear and extending tool life [8]. The research by Lazarus et al. [9] found that highly chlorinated cutting fluid can effectively improve the cutting performance when cutting tantalum–tungsten alloy Ta-2.5W. This is attributed to the characteristics of the cutting fluid, which can prevent the formation of built-up edges, reducing adverse effects during the cutting process and making machining more efficient. However, due to the potential environmental and health hazards associated with the excessive use of cutting fluids, numerous scholars propose alternative methods, such as utilizing low-temperature CO_2_ and liquid nitrogen as substitutes for cutting fluids. A study conducted by Wang et al. [10] employed a combination of traditional turning and cryogenic machining for tantalum processing. The experimental results demonstrated that this method could significantly reduce surface roughness by 200%, decrease cutting forces by 60%, and extend tool life by 300%. Furthermore, another investigation by Wang et al. [11] delved into the impact of liquid nitrogen cooling and MQL + CO_2_ cooling on tool wear and chip formation during the cutting of tantalum–tungsten alloy. The results indicated that under these two low-temperature cooling conditions, tool life could be improved by 50%, significantly reducing adhesive wear. Tool failure primarily manifested as notch wear and flank wear, while there was also a substantial reduction in chip thickness and cutting forces.

In addition to exploring the influence of cutting fluids on the machining performance of tantalum–tungsten alloy, tool parameters are an integral component. In metal cutting processes, a well-crafted tool structure serves as the cornerstone for maximizing the cutting performance of the tool [12]. Mizutani et al. [13], with the goal of minimizing tool wear in pure tantalum cutting, employed an electrolytic in-process dressing (ELID) grinding system to fabricate tools with rake angles of 23°, 28°, and 33°. Experimental assessments of these three rake angles revealed that the carbide tool with a 28° rake angle exhibited the lowest cutting forces and the highest tool life, rendering it particularly suitable for the precision machining of tantalum materials. Furthermore, Lazarus [9] utilized a single carbide end mill to machine tantalum–tungsten alloy material at a cutting speed of 74.7 m/min. The study demonstrated that a fine-grained carbide end mill could extend tool life significantly. For turning operations, it is recommended that the rake angle of carbide tools should not be less than 20°. While a larger rake angle can enhance the machining performance of tantalum–tungsten alloy, it may compromise the cutting stability of the tool. The tool life exhibits substantial fluctuations, ranging from 1 to 9.75 min.

Concurrently, the impact of coating materials on the machining performance of tantalum–tungsten alloy is paramount. John et al. [14] undertook machining of tantalum–tungsten alloy using nano-coatings in conjunction with tungsten carbide cutting inserts. They compared the influence of coated and uncoated tools on the cutting process of tantalum–tungsten alloy. The study revealed that titanium-based coatings could extend tool life, with the predominant wear mechanisms observed as notch wear and fracture wear. In a study conducted by Wang et al. [15], the tool life and wear characteristics of coated and uncoated tools during the turning of tungsten heavy alloy were examined. TiAlN-coated, TiCN-coated, and uncoated tools were employed for turning operations. The results demonstrated that TiAlN-coated tools exhibited the longest tool life and superior cutting performance. Pritima et al. [16] conducted a study comparing the impact of three coating materials—Al_2_O_3_, TiAlN, and TiC—on tool wear and cutting temperature during the turning process of high-silicon and aluminum alloy. The research findings indicate that using TiC-coated tools with a coating thickness of 3 μm leads to a 34% lower cutting temperature compared to other tools. Additionally, there is a slight reduction in the tool wear rate. Furthermore, Li et al. [17] investigated the wear mechanisms and machining surface integrity of AlTiN-coated tools during the liquid nitrogen cutting of GH605 cobalt-based high-temperature alloy. The findings indicated that by reducing the liquid nitrogen injection temperature, severe tool wear could be effectively delayed, side damage minimized, and oxidation wear inhibited. Jadam et al. [18] conducted a study on the cutting performance of PVD multilayer-coated metal ceramic (TiN/TiCN/TiN) and PVD-coated (TiAlN) cemented carbide tools when machining Inconel 718 alloy. The results indicated that the coated metal ceramic tools exhibited lower cutting forces, superior surface quality, and reduced wear on the rake face compared to the PVD-coated cemented carbide tools. Thus, coating materials not only diminish tool wear and enhance tool life but also serve as an effective means to significantly improve tool performance [19,20].

Based on the literature review, current research on coating materials primarily concentrates on challenging-to-machine alloys such as tungsten alloys and nickel-based high-temperature alloys. However, there is a noticeable gap in the research regarding the cutting mechanisms of tantalum–tungsten alloy concerning coating materials. Given that the wear mechanism of cutting tools is significantly influenced by coating materials, especially during high-speed cutting of Ta-2.5W, selecting appropriate coating materials is crucial to slowing down the rate of tool wear. In this study, Ta-2.5W serves as the workpiece material, and turning experiments are conducted using uncoated cemented carbide tools, TiAlN-coated cemented carbide tools, and AlTiN-coated cemented carbide tools. The aim is to investigate the impact of coating materials on the cutting tool’s surface temperature, the morphology of the tool’s surface wear, and the characteristics of chip formation. This research endeavors to provide essential theoretical support and a technical basis for improving efficiency in the turning process of tantalum–tungsten alloy and slowing down the rate of tool wear.

## 2. Materials and Methods

### 2.1. Workpiece and Tool Materials

This experimental work utilized tantalum–tungsten alloy Ta-2.5W as the workpiece material, characterized by a hardness of 130 HV. The workpiece dimensions were 400 mm in length and 40 mm in diameter. The chemical composition and mechanical properties of the Ta-2.5W tantalum–tungsten alloy rod material are presented in Table 1. This material and its corresponding chemical composition table were provided by Changsha Southern Tantalum-Niobium Co., Ltd. (Changsha, China). The cutting tools used in the experiment were sourced from Xiamen Jinlu Special Alloy Co., Ltd. (Xiamen, China), and they include uncoated carbide tools, TiAlN-coated carbide tools, and AlTiN-coated carbide tools. The substrate is WC-Co-based carbide alloy, and the coating thickness is 3 μm. The tool specifications feature a 15° rake angle, a 7° relief angle, a 0.2 mm corner radius, with DCMT11T302 insert type, and an SDJCR2020K11 type toolholder. Detailed parameters for the cutting inserts are provided in Table 2 [11].

### 2.2. Cutting Experiment

The machining experiment of tantalum–tungsten alloy Ta-2.5W was conducted on the CK6140S CNC lathe, as illustrated in Figure 1, depicting the experimental setup, workpiece material, and their assembly relationships. The cutting parameters required for the experiment are listed in Table 3. To account for experimental errors, each experiment was repeated three times.

In the cutting experiments, axial turning was conducted with intervals of 100 mm for measuring tool wear, utilizing the VHX-7100 Ultra Depth 3D Digital Microscope manufactured by Keyence Corporation, Japan. Due to the irregular nature of wear on the rear tool face, the wear area underwent three measurements, and the average value was recorded on the wear curve. According to the International Organization for Standardization (ISO 3685) [21], a wear standard of VB = 0.3 mm on the rear tool face is applied. When the wear on the rear tool face failed to meet the failure criterion after cutting a length of 100 mm, or if there was no noticeable chipping on the front tool face, to avoid experimental errors arising from intermittent cutting, the tool was replaced and a new tool was utilized for continuous cutting of a 200 mm length. Following the cutting process, the tool was removed again, and both the front and rear tool faces were observed using the digital microscope to ascertain tool failure. If the tool had not failed, the experiment was repeated following the steps until failure occurred. Upon tool failure, the cutting length and wear on the rear tool face at the point of failure were recorded to facilitate data collection for calculating tool life. Tool life is determined based on cutting time [11], and given that the workpiece diameter changes with each cut, thereby affecting spindle speed, to mitigate this error, constant linear velocity cutting was adopted on the lathe. The workpiece diameter was recorded every 100 mm of cutting, and this diameter served as the input for the next 100 mm of cutting in Equation (1) to calculate the cutting time.
(1)$$t=\frac{60\pi DL}{1000vf}$$

In the equation, *t* represents the cutting time (s), *D* is the workpiece diameter (mm), *L* is the cutting length (mm), *v* is the cutting speed (m/min), and *f* is the feed rate (mm/rpm).

The cutting surface temperature was measured using the DT-988H infrared thermal imager produced by Huashengchang Company (Shenzhen, China). To ensure the accuracy of the measurement results, it was essential to maintain consistency in the measurement distance, time intervals, and area. To avoid experimental errors, the infrared thermal imaging camera needs to be calibrated before use. Additionally, a scanning electron microscope (SEM) was employed to observe the worn morphology and chip morphology on the tool surface after cutting. The elemental distribution in the worn area was analyzed using an energy-dispersive X-ray spectrometer (EDS). The experimental instruments used for measurements are depicted in Figure 2.

## 3. Results and Discussion

### 3.1. Cutting Temperature

Figure 3 depicts the temperature contour maps obtained through temperature measurements in the cutting zone for AlTiN-coated tools under various cutting conditions: *V_c_* = 50 m/min; *a_p_* = 0.1 mm, *f_z_* = 0.06 mm/rpm; *V_c_* = 100 m/min, *a_p_* = 0.1 mm, *f_z_* = 0.18 mm/rpm; and *V_c_* = 150 m/min, *a_p_* = 0.1 mm, *f_z_* = 0.12 mm/rpm. From Figure 3, it is evident that the higher temperature regions are primarily concentrated at the contact position between the tool tip and the workpiece, with the temperature field exhibiting a staircase-like spread outward. In Table 4, the orthogonal experimental results for cutting temperature under different cutting parameters are presented for the selected tools. Both the AlTiN and TiAlN coatings prove effective in reducing the cutting surface temperature, with the TiAlN coating exhibiting a more significant impact. This is attributed to the outstanding thermal conductivity of the TiAlN coating, facilitating the rapid transfer of heat generated during cutting and efficiently conducting heat from the tool surface to the base material, thereby lowering the surface temperature. Moreover, the TiAlN coating demonstrates exceptional chemical stability at high temperatures, resisting oxidation and chemical reactions, and consequently retarding the thermal degradation of the coating. In contrast, the AlTiN coating may be more susceptible to oxidation in high-temperature and high-friction environments, leading to a decline in coating performance and an increase in surface temperature. Additionally, TiAlN coatings typically feature lower friction coefficients, reducing friction between the tool and the workpiece, which aids in slowing down the temperature rise. Conversely, the AlTiN coating has a relatively higher friction coefficient, potentially generating more heat from friction. This combination of effects contributes to an overall enhancement in the efficiency of the cutting process. Moving forward, the Taguchi analysis method was applied to scrutinize the factors influencing tool cutting temperature, and detailed results are presented in Table 5.

Delta reflects the extent of these factors’ impact on the cutting temperature, with a larger value indicating a greater influence of the corresponding cutting parameters. It is observed that cutting speed has the most significant effect on the cutting temperature of this alloy, followed by cutting depth, while the feed rate has the least impact. The influence of cutting parameters on cutting temperature is illustrated in Figure 4. It is evident that cutting temperature sharply increases with the rise in cutting speed, and the rate of temperature increase is relatively slower with increasing cutting depth. The feed rate has a minimal effect on cutting temperature.

### 3.2. Tool Life

Cutting speed significantly influences tool wear. Generally, higher cutting speeds result in shorter tool life. However, slower cutting speeds may impact processing efficiency and potentially be detrimental to tool life [15]. Therefore, in this study, cutting time (in seconds) is used to represent tool life. To ensure the quality of the processed product, a criterion of VB = 0.3 mm for tool wear on the rear rake face is adopted as the failure standard. If the wear on the rear rake face exceeds 0.3 mm after the final cut, the tool life is recorded as the value at the VB = 0.3 mm point, obtained by fitting the tool wear and cutting time curve. Figure 5 illustrates the tool life of the AlTiN-coated, TiAlN-coated, and uncoated tools under different cutting parameters. It can be observed that tool life initially increases and then decreases with increasing cutting speed, cutting depth, and feed rate. Additionally, the coated carbide tools exhibit longer tool life compared to the uncoated tools, with the TiAlN-coated carbide tools showing the longest tool life. When the cutting speed is 100 m/min, cutting depth is 0.2 mm, and feed rate is 0.12 mm/rpm, the TiAlN-coated carbide tool achieves the longest tool life at 457.74 s.

Figure 6 depicts the wear progression of three types of tools as the cutting length increases under the machining conditions of *V_c_* = 100 m/min; *a_p_* = 0.3 mm; and *f_z_* = 0.06 mm/rpm. With the growth in the axial cutting distance, the width of wear on the rake face gradually expands. In comparison to the coated tools, the uncoated tool experiences a faster wear rate, exhibiting an approximately linear increase in tool wear. By the time the axial cutting distance reaches 183.58 mm, the wear on the rake face of the uncoated tool has already surpassed the failure criterion of 0.3 mm. This is attributed to the absence of a protective layer on uncoated tools, which fails to mitigate tool wear during turning processes [22]. Moreover, upon closer examination of Figure 6, it is evident that the coated tools demonstrate significantly longer lifespans than the uncoated tool, particularly the TiAlN-coated carbide tool. This can be attributed to the high hardness of both coatings, effectively enhancing the overall hardness of the tools, thereby reducing the wear rate and extending tool life. Additionally, the TiAlN coating exhibits higher temperature stability compared to the AlTiN coating, making it more suitable for high-temperature cutting conditions. Considering the alloy generates elevated temperatures during cutting, the TiAlN coating can maintain good performance in high-temperature environments, assisting in mitigating the thermal wear of the tool.

### 3.3. Tool Wear

Tool wear is a critical quality aspect in machining operations, significantly impacting cutting edge geometry, machine surface quality, workpiece dimensions, and cutting mechanisms [23]. Under cutting conditions with *V_c_* = 100 m/min; *a_p_* = 0.3 mm; and *f_z_* = 0.06 mm/rpm, we employed a digital microscope to capture images of the rear and front tool faces of AlTiN-coated, TiAlN-coated, and uncoated tools to observe wear morphology during the cutting process, as depicted in Figure 7. In Figure 7a, it is evident that the initial wear pattern on the rear tool face of the AlTiN-coated tool resembles a W-shape, gradually transitioning into a U-shape with increased cutting length. This transition results from the influence of cutting forces and heat on the tool, leading to an uneven distribution of local temperature and stress on the tool surface. This non-uniform distribution makes specific areas of the tool surface more susceptible to wear, forming W-shaped depressions. Moreover, chip flow during cutting along the rear tool face contributes to impact and friction on the tool surface. Different coatings may impact chip flow differently, causing more pronounced wear in specific areas, forming U-shaped depressions. As seen in Figure 7c, the rear tool face of the TiAlN-coated tool displays a similar wear pattern, with size gradually increasing over continuous cutting. Observing Figure 7e, the rear tool face of the uncoated tool also exhibits a W-shaped wear pattern, but with smaller U-shaped wear.

Based on the observations from Figure 7b, it can be concluded that at a cutting length of L = 100 mm, the cutting edge of the AlTiN-coated tool exhibits slight damage, and the coating begins to delaminate. With the increase in cutting length, the damage to the cutting edge further expands, and significant wear is observed on the primary cutting edge, with the delamination area of the coating gradually increasing. This might occur due to the gradual reduction in adhesion between the coating and substrate under elevated temperature and pressure conditions. Additionally, fluctuations in cutting forces and temperatures, along with the influence of the workpiece material, contribute to a decrease in coating stability, leading to an expansion of the delaminated coating area. Up to a cutting length of L = 300 mm, with a rear flank wear amount of VB = 326.15 µm, tool failure occurs, accompanied by the occurrence of edge chipping. Crescent-shaped wear can still be observed on the rake face. This may be related to specific mechanical effects during the coating delamination and cutting process. Observing Figure 7d, for the TiAlN-coated tool, in the initial stage of cutting, the cutting edge also experiences slight wear, and the coating on the rake face begins to delaminate. With the increase in cutting length, the wear of the cutting edge continues to expand, and chip residue appears on the rake face. When the cutting length reaches L = 400 mm, with a rear flank wear amount of VB = 303.18 µm, the tool reaches the failure criterion, and edge chipping occurs simultaneously. A significant scratch can also be observed on the rake face, possibly due to the pronounced marks left on the rake face of the tool due to coating delamination. Overall, the U-shaped wear pattern on the rear flank and delamination and scratching phenomena on the rake face of the TiAlN-coated tool indicate a significant influence of the coating on tool performance and life, particularly under high-temperature and high-friction conditions, where the coating can more effectively protect the tool surface. From Figure 7f, in the initial stage of cutting, the primary cutting edge of the uncoated tool has already experienced slight edge chipping. When the cutting length reaches L = 200 mm, with a rear flank wear amount of VB = 160.21 µm, the failure criterion for the rear flank has not yet been reached, but edge chipping has already occurred, with most of the primary cutting edge lost, thus determining that the tool has failed at this point. This indicates that the absence of a coating plays an important role in tool failure.

To provide a clearer depiction of the wear mechanisms on the front surface of the AlTiN-coated and TiAlN-coated tools, Figure 8 presents the energy-dispersive X-ray spectroscopy (EDS) line scan analysis results of these two tool types when they fail under cutting conditions of *V_c_* = 100 m/min; *a_p_* = 0.3 mm; and *f_z_* = 0.06 mm/rpm. Upon examining the Figure 8, it becomes apparent that regions closer to the tool tip exhibit a substantial presence of C elements and lesser amounts of Ti, Al, and N elements, indicating severe coating delamination with substrate exposure. Furthermore, significant levels of Ta and W elements are detected in the tool tip region, suggesting that after coating delamination, workpiece adhesion occurs on the tool’s front surface, forming chip accumulation. This leads to adhesive wear, exacerbating tool wear and subsequently reducing tool life. Additionally, in the scanning region from point A to point B, a notable amount of O element is detected, signifying an oxidation reaction between the coating material and oxygen, resulting in the formation of an oxide that may flow out with the chip, causing tool oxidation wear. Furthermore, even in wear regions far from the tool tip, traces of Ta, W, and O elements persist, indicating diffusion wear on the tool. This implies that during the cutting process, the tool undergoes various wear mechanisms, including adhesive wear, diffusion wear, and oxidation wear.

Figure 9 displays the elemental content of the rake face of the two types of coated tools after data processing. The analysis of the results reveals that adhesive and oxidative wear on the rake face are more pronounced in the case of the AlTiN-coated tool compared to the TiAlN-coated one. This might be attributed to the relatively higher instability of the AlTiN coating under conditions of elevated temperature and friction, making it susceptible to oxidation reactions, thereby resulting in coating delamination and surface oxidation of the tool. In contrast, the TiAlN coating demonstrates better stability under similar conditions, effectively mitigating adhesive and oxidative wear, thereby enhancing the wear resistance and lifespan of the tool.

## 4. Conclusions

This paper outlines the machining of Ta-2.5W using coated tools and examines the cutting temperature, tool lifespan, and tool wear characteristics of TiAlN-coated carbide tools, AlTiN-coated carbide tools, and uncoated carbide tools under various cutting parameter combinations during the turning process of Ta-2.5W. The conclusions drawn from the tool performance on this alloy include the following:

1. The cutting speed has the most pronounced impact on cutting temperature in Ta-2.5W alloy, followed by cutting depth, with feed rate having the least effect. Additionally, cutting temperature sharply increases with rising cutting speed, while the rate of temperature increase relative to cutting depth is comparatively slower. The feed rate exhibits minimal influence on cutting temperature. Under identical cutting conditions, the coated tools exhibit a significant advantage over the uncoated carbide tools. These coated tools notably reduce the cutting surface temperature, minimize tool wear, and significantly enhance tool lifespan, with the TiAlN coating demonstrating particularly remarkable effectiveness. Compared to the uncoated tools, the TiAlN-coated tools can reduce cutting temperatures by approximately 6% to 21%. Similarly, when compared to the AlTiN-coated tools, the cutting temperature is reduced by approximately 0.7% to 5%.

2. In turning experiments, the width of wear on the back face gradually increases with the axial cutting distance. The uncoated tools display a faster wear rate, with tool wear increasing almost linearly. At an axial cutting distance of 183.58 mm, the wear on the back face of the uncoated tools has reached the failure criterion of 0.3 mm. Under similar Ta-2.5W turning conditions, the tool lifespan of the coated carbide tools exceeds that of the uncoated carbide tools, with the TiAlN-coated carbide tools exhibiting the longest lifespan. Compared to the uncoated tools, the TiAlN-coated tools can enhance tool life by approximately 117%. Meanwhile, compared to the AlTiN-coated tools, the increase in tool life is approximately 41%.

3. During the cutting process, the back face represents the primary wear form of the turning tools, displaying a wear pattern transitioning from W-shaped to U-shaped with increasing cutting length. The front face also undergoes adhesive wear, oxidation wear, and diffusion wear. Among these three types of tools, the wear on the TiAlN-coated carbide tools is notably less severe compared to that on the AlTiN-coated carbide tools and the uncoated carbide tools. TiAlN-coated carbide tools are poised to be suitable for cutting Ta-2.5W.

## Figures and Tables

**Figure 1 materials-17-01481-f001:**
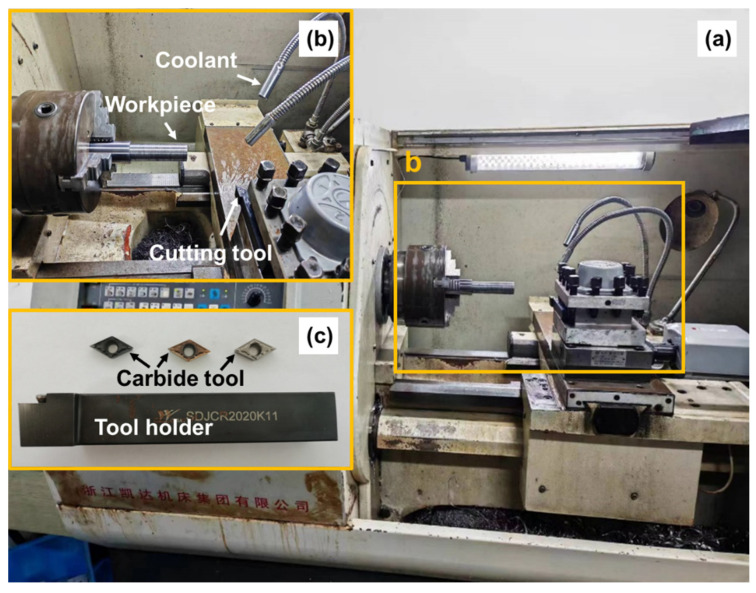
Experimental setup employed in turning procedures: (**a**) CNC lathe, (**b**) assembly relationship between workpiece and tool, (**c**) tool holder with cutting inserts.

**Figure 2 materials-17-01481-f002:**
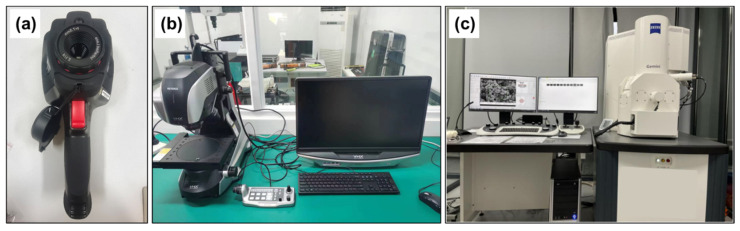
Main measurement instruments: (**a**) infrared thermal imager, (**b**) super-depth-of-field 3D digital microscope, (**c**) Zeiss sigma 300 scanning electron microscope.

**Figure 3 materials-17-01481-f003:**
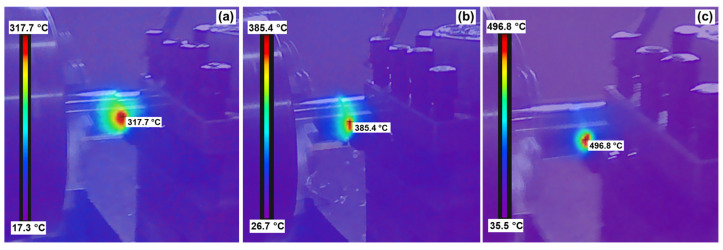
Cutting temperature contour map: (**a**) *V_c_* = 50 m/min, *a_p_* = 0.1 mm, *f_z_* = 0.06 mm/rpm; (**b**) *V_c_* = 100 m/min, *a_p_* = 0.1 mm, *f_z_* = 0.18 mm/rpm; (**c**) *V_c_* = 150 m/min, *a_p_* = 0.1 mm, *f_z_* = 0.12 mm/rpm.

**Figure 4 materials-17-01481-f004:**
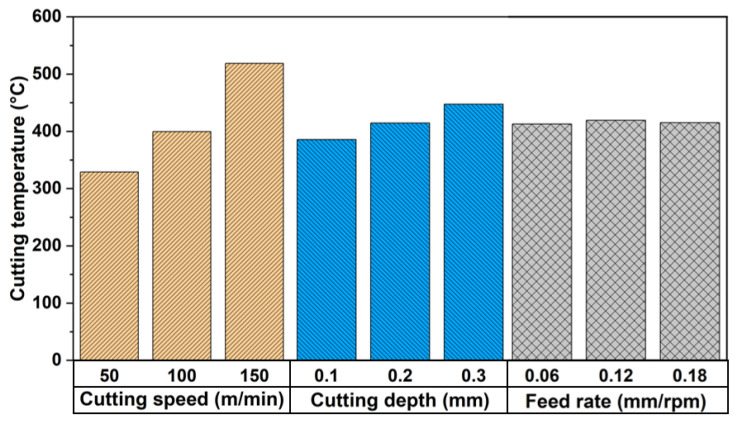
Influence of cutting parameters on cutting temperature for AlTiN-coated cemented carbide cutting tools.

**Figure 5 materials-17-01481-f005:**
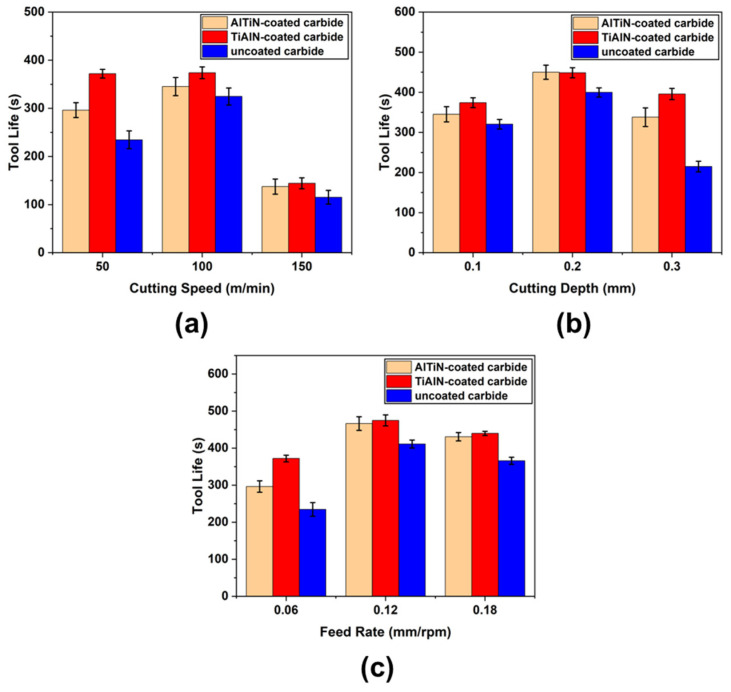
Tool lives under different tool materials and cutting parameters: (**a**) different cutting speeds, (**b**) different cutting depths, (**c**) different feed rates.

**Figure 6 materials-17-01481-f006:**
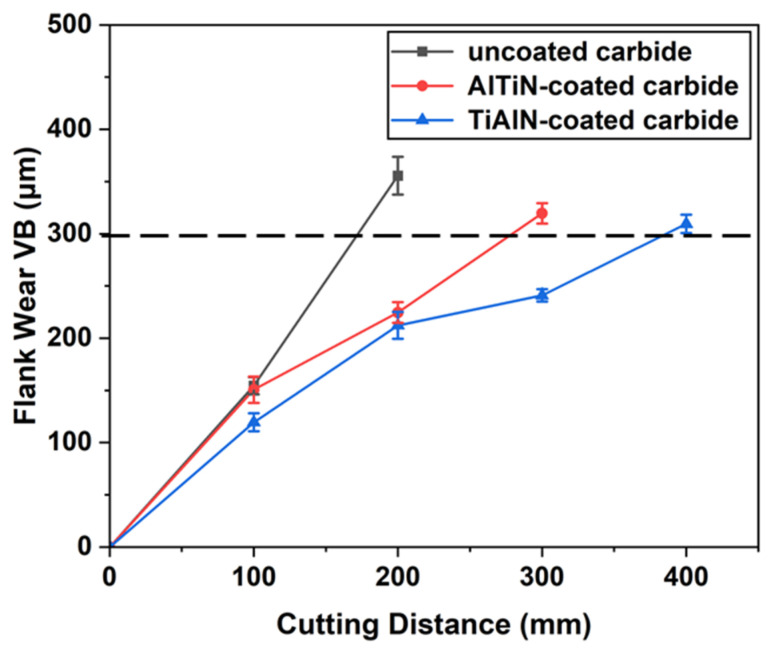
Tool wear progression under different tool materials.

**Figure 7 materials-17-01481-f007:**
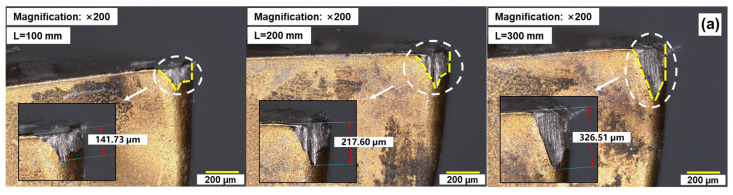
Tool surface wear images: (**a**) AlTiN-coated tool rear face, (**b**) AlTiN-coated tool rake face, (**c**) TiAlN-coated tool rear face, (**d**) TiAlN-coated tool rake face, (**e**) uncoated tool rear face, (**f**) uncoated tool rake face.

**Figure 8 materials-17-01481-f008:**
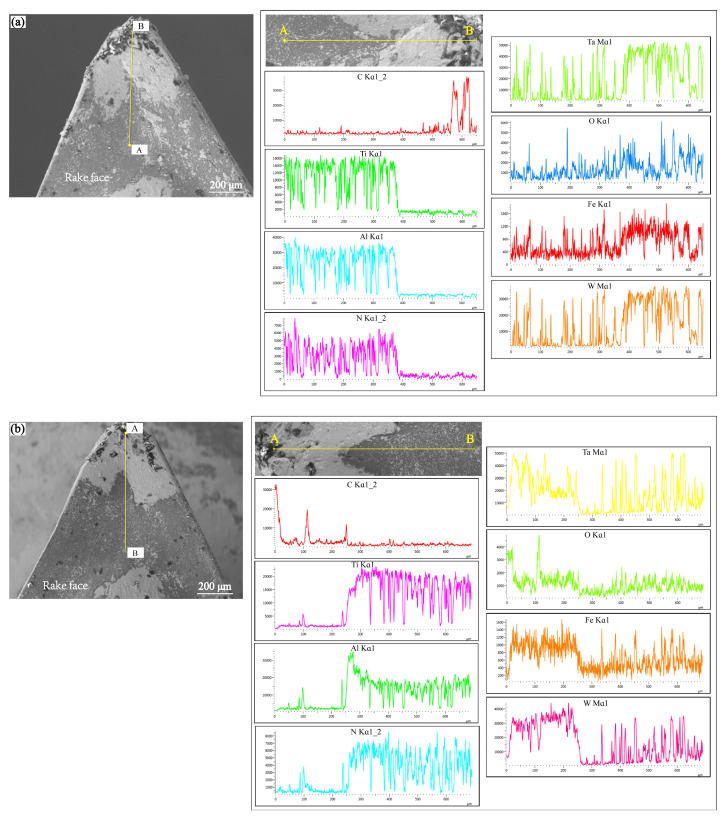
EDS results along lines from A to B for the cutting tool: (**a**) AlTiN-coated carbide, (**b**) TiAlN-coated carbide.

**Figure 9 materials-17-01481-f009:**
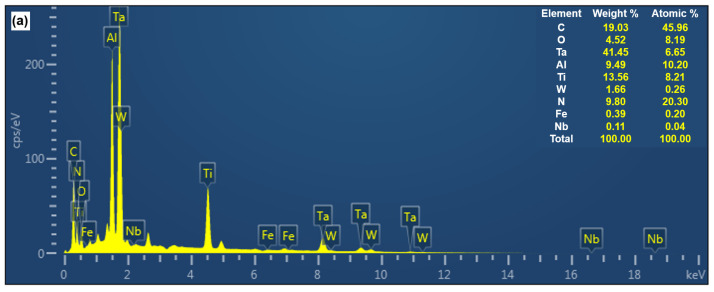
EDS result of rake face element distributions: (**a**) AlTiN-coated carbide, (**b**) TiAlN-coated carbide.

**Table 1 materials-17-01481-t001:** Chemical composition and mechanical properties of Ta-2.5W at room temperature.

Chemical Composition	Ratio (wt.%)	Chemical Composition	Ratio (wt.%)	Mechanical Property	Value
Ta	Main element	O	≤0.015	Yield Strength σs	286 MPa
W	≤2.75	Nb	≤0.500	Elastic Modulus E	149 GPa
C	≤0.010	Fe	≤0.010	Hardness	130 HV
N	≤0.010	Ti	≤0.010	Elongation	20%
H	≤0.0015			Shrinkage	40%

**Table 2 materials-17-01481-t002:** Specific parameters of cutting inserts.

Cutting Tools: DCMT11T302				
Rake angle ($\gamma_{0}$)	Clearance angle ($\alpha_{0}$)	Corner radius ($r_{\varepsilon }$)	Length L	Thickness S
15°	7°	0.4 mm	11.6 mm	3.97 mm

**Table 3 materials-17-01481-t003:** Orthogonal experimental table of cutting parameters.

Group	Cutting Speed (m/min)	Cutting Depth (mm)	Feed Rate (mm/rpm)
1	50	0.1	0.06
2	100	0.2	0.12
3	150	0.3	0.18

**Table 4 materials-17-01481-t004:** Cutting temperature test results.

Cutting Speed (m/min)	Cutting Depth (mm)	Feed Rate (mm/rpm)	AlTiN-Coated Tool Cutting Temperature (°C)	TiAlN-Coated Tool Cutting Temperature (°C)	Uncoated Tool Cutting Temperature (°C)
50	0.1	0.06	304.8	289.7	368.5
50	0.2	0.18	336.5	328.4	389.4
50	0.3	0.12	379.6	368.9	428.6
100	0.1	0.18	385.4	373.4	468.5
100	0.2	0.12	407.5	395.5	482.7
100	0.3	0.06	439.5	429.4	501.3
150	0.1	0.12	496.8	493.2	523.4
150	0.2	0.06	528.8	519.2	558.6
150	0.3	0.18	564.4	543.7	594.1

**Table 5 materials-17-01481-t005:** Analysis results.

	Cutting Temperature (°C)		
	Cutting Speed	Cutting Depth	Feed Rate
1	329.0	385.5	412.8
2	399.4	414.4	419.2
3	518.8	447.4	415.2
Delta	189.7	61.9	6.4

## Data Availability

Data are contained within the article.

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
