# Peer review of "Research on Coated Tool Life and Wear in Ta-2.5W Alloy Turning"

_materials, 2024, doi:10.3390/ma17071481_

Round 1
Reviewer 1 Report
Comments and Suggestions for Authors
This is a good work on tool life while cutting a difficult alloy. Lot of different testing techniques were used in this work but I realised there are some issues that need to be addressed:
1) The title is not clear at all, i had misunderstood thinking that was a study on turning tools made of Ta-2.5W.
2) Some adjectives when used for this material are not ideal: ex very viscous.
3) On line 82, reducing the injection temperature of liquid nitrogen? Not sure what this means.
4) A picture of the tool, with dimensions and possibly details on the coatings would be really helpful.
5) VHX-7100 microscope, who is the producer? Keyence? if so, say it.
6) Tool wear could be assessed by measuring the average wear, or by assessing the total volume loss. The latter would be much more effective.
7) Infrared measurements taken with a camera are rather imprecise as the emissivity of a moving piece can be tricky to be calculated.
8) Table 4 has a mispelt AlTiN (AiTiN) and formatting is otherwise not in shape for publishing.
9) Why measuring tool life by time when you can use a normalised unit like material removal rate? I do not see the point.
10) All histograms and graphs should include errors or STD.
11) There is no clear indication on how many tests were performed. Single tests for each insert would result in statistically unsound results.
12) I see a significant amount of delamination on the tools.
13) Conclusions: point 1 cannot really be defended, as the thermal camera is quite an imprecise tool. point 2 is trivial. Remains only point 3, which is not the strongest one...these conclusions are a let down the way they are written.
I will recommend a revision because this is good work but not presented in the appropriate way. If only ONE test per tool was performed, I would recommend re-submission after a sufficient statistic was gathered.
Comments on the Quality of English Language
English is quite good in general. Some wrong adjectives and a couple of too long sentences were spotted. Nothing serious.
Reviewer 2 Report
Comments and Suggestions for Authors
It is a very interesting paper with relevant results. I've localized some minor changes that would be necessary to correct before acceptance, as follows.
- There is an error in the title of Table 1 (it reads: Table 1. This is a table. Tables should be placed in the main text near to the first time they 113 are cited)
- Figure 1 seems to not bring any relevant information to the manuscript.
- The measurements do not show their uncertainty. It would be really relevant, especially in the bar graphs in which the difference may not seem very large.
Reviewer 3 Report
Comments and Suggestions for Authors
The paper deals with cutting operation of Ta-2.5W using three types of cemented carbide tools: uncoated, TiAlN-coated, and AlTiN-coated. The authors investigate the impact of coating materials on the cutting tool's surface temperature, the morphology of tool surface wear, and the characteristics of chip formation. The paper is interesting and well-written. Only few revisions are necessary.
Line 50: CO2. Please put 2 subscript
Line 104: is not necessary to write initial
Line 107: Please, if you know, add the thickness and other specifications of the coatings.
Line 113: Please delete “This is a table” and “Tables should be placed in the main text near to the first time they are cited” and write the correct caption.
Figure 3. Please specify which is the cutting tool.
Comments on the Quality of English LanguageEnglish is quite good
Reviewer 4 Report
Comments and Suggestions for Authors
1. Please explain in the text what the abbreviation ELID means. Line 59
2. Description of table 1 to be improved. Line 113
3. How the data from tables 1 and 2 were determined or measured. Please describe in the text or provide a literature reference
4. Line 119 "The cutting parameters required for the experiment, as derived from literature sources" - please specify from which literature sources the parameters were selected (manufacturer's catalog, article, etc.)
5. Marking a in Figure 5 is illegible
6. In your application, please refer to the numerical values obtained during the tests, e.g. temperatures, tool wear, comparison of tool operating times,
7. Whether and how the thermal imaging camera was calibrated
Round 2
Reviewer 1 Report
Comments and Suggestions for Authors
After reviewing the resubmitted work I was rather surprised to find still some of the points that I raised in the first instance:
1) Hard alloy = I guess the authors mean tungsten carbide cutting tools, but it is not defined clearly, as it is, it could be misunderstood for HSS.
2) Thermal camera images have a weak correlation and I would have really liked the authors to perform a simple calibration first by assessing the emissivity for their workpiece at the expected temperatures.
3) What does it mean "rear tool face wear amount"? The tool wears on the rake face, not on the rear face..
4) Coating delaminates, it doesn't peel. Please use appropriate technical terms.
This work is almost in a ready state for publication, even if significance remains low. I would recommend still to correct the points I have highlighted, if all "hard alloy" is replaced with carbide, misunderstanding can be kept to a minimum.
Comments on the Quality of English Language
Hard alloy - peeling
These words are repeated several times in this work, they should be replaced by the appropriate technical terms.
Reviewer 4 Report
Comments and Suggestions for Authors
1. In the descriptions of tables 1 and 2 will be added by literature references
